# Peptidomics Analysis Reveals the Buccal Gland of Jawless Vertebrate Lamprey as a Source of Multiple Bioactive Peptides

**DOI:** 10.3390/md21070389

**Published:** 2023-06-29

**Authors:** Yaocen Wang, Feng Sun, Zhuoying Wang, Xuyuan Duan, Qingwei Li, Yue Pang, Meng Gou

**Affiliations:** 1College of Life Science, Liaoning Normal University, Dalian 116081, China; wang_yaocen@163.com (Y.W.); sunfeng6085@163.com (F.S.); wangzhuoying0306@163.com (Z.W.); duanxuyvan@163.com (X.D.); liqw@263.net (Q.L.); 2Lamprey Research Center, Liaoning Normal University, Dalian 116081, China; 3Collaborative Innovation Center of Seafood Deep Processing, Dalian Polytechnic University, Dalian 116034, China

**Keywords:** lamprey, peptidomics, antimicrobial, anti-inflammatory, endogenous peptides

## Abstract

Various proteins with antibacterial, anticoagulant, and anti-inflammatory properties have been identified in the buccal glands of jawless blood-sucking vertebrate lampreys. However, studies on endogenous peptides in the buccal gland of lampreys are limited. In this study, 4528 endogenous peptides were identified from 1224 precursor proteins using peptidomics and screened for bioactivity in the buccal glands of the lamprey, *Lethenteron camtschaticum*. We synthesized four candidate bioactive peptides (VSLNLPYSVVRGEQFVVQA, DIPVPEVPILE, VVQLPPVVLGTFG, and VPPPPLVLPPASVK), calculated their secondary structures, and validated their bioactivity. The results showed that the peptide VSLNLPYSVVRGEQFVVQA possessed anti-inflammatory activity, which significantly increased the expression of anti-inflammatory factors and decreased the expression of inflammatory factors in THP-1 cells. The peptide VVQLPPVVLGTFG showed antibacterial activity against some gram-positive bacteria. The peptide VSLNLPYSVVRGEQFVQA possessed good ACE inhibitory activity at low concentrations, but no dose-related correlation was observed. Our study revealed that the buccal glands of the jawless vertebrate lamprey are a source of multiple bioactive peptides, which will provide new insights into the blood-sucking mechanism of lamprey.

## 1. Introduction

Owing to their special feeding mechanisms, the salivary glands of blood-sucking animals are considered an important source of bioactive proteins and peptides. The representative bioactive peptide of the oral glands of blood-sucking animals is called hirudin, a polypeptide composed of 64–66 amino acids with a molecular weight of approximately 7000 Da, which is considered the strongest natural and specific inhibitor of thrombin found so far [1]. In addition, various bioactive peptides have been identified in the salivary glands of blood-sucking animals. For example, various antimicrobial peptides (ixosin-B, ixosin, ISAMP, Rhamp, and longicornsin) [2,3,4,5,6] and an immunosuppressant peptide (amregulin) [7] have been identified in the salivary glands of ticks, which help ticks overcome the host’s defense response during sucking blood and maintain the blood in a sterile condition. Vasodilator peptides (vasotab and vasotab TY) [8,9], immunosuppressive peptides (tabimuregulins), and antimicrobial peptides (attactin, defensing, and cecropin) [10] have been identified in the salivary glands of horse flies. AeMOPE-1, a peptide found in *Aedes aegypti*, exhibits selective immunomodulatory activity against macrophages [11]. In addition to hirudin, the salivary glands of leeches secrete peptides with immunomodulatory and antihemostatic functions, such as neuropeptide (neuropeptide Y-HS), which can suppress the inflammatory response of the host [1,12].

Lampreys are primitive jawless vertebrates, most species of which undergo complete metamorphosis from filter-feeding larvae (ammocoetes) to parasitic juveniles that feed on the blood of the host fish [13]. *Lethenteron camtschaticum* is a large species of lamprey distributed in Russia, Japan, and China, with dark brown secretions in its buccal glands that are squeezed into the mouth to assist in blood sucking during the parasitic process [14]. Several functional proteins in the buccal glands of *L. camtschaticum* have been identified and characterized, such as buccal gland secretion protein-1 (BGSP-1), which can prevent blood coagulation in the host fish, and the RGD (Arg-Gly-Asp) motif toxin in the lamprey buccal gland, which can directly cooperate with BGSP-1 to act as an anticoagulant [15]. A member of the cysteine-rich secretory protein family, cysteine-rich buccal gland protein (CRBGP), may help lampreys inhibit vasoconstriction and the nociceptive responses in host fish during blood sucking and prevent lampreys from being injured by reactive oxygen species (ROS) [16,17,18]. The translationally controlled tumor protein (TCTP) from the buccal gland of *L. camtschaticum* inhibits the coagulation cascade, induces vasodilation in host fish during blood sucking, and participates in immune regulation between the lamprey and host fish [19]. Although much is known about the bioactive proteins in the buccal glands of lamprey, research on peptides in the buccal glands remains limited.

High-resolution mass spectrometry (MS)-based proteomics is a common strategy used for proteomic research in biology. However, in most proteomic studies, endogenous peptides cannot be distinguished from enzymatically cleaved peptides, as samples require enzymatic cleavage, and potentially bioactive endogenous peptides are overlooked [20,21]. Peptidomics focuses on the comprehensive characterization of peptides of biological origin using MS. Although peptidomics is closely linked to proteomics, it differs fundamentally from conventional bottom-up proteomics [21]. Peptidomics is now being used more widely in bioactive peptide and biomarker discovery [22,23,24].

This study aimed to explore or characterize bioactive peptides from the buccal gland of jawless vertebrate lamprey. Secretion from the buccal gland of lamprey was fractionated and characterized using peptidomic analysis. Four screened candidate peptides were synthesized, and their biological functions were validated. Our study revealed that the buccal gland of lamprey is a potential source of multiple bioactive peptides, which will improve our understanding of the molecular interactions that occur between lamprey and host fish and will provide new insights regarding the blood-sucking mechanism of lampreys.

## 2. Results

### 2.1. Identification of Endogenous Peptides from the Buccal Gland of Lamprey Using Peptidomics Analysis and Bioinformatics Analysis

MS analysis detected 4528 endogenous peptides from 1224 precursor proteins derived from the buccal gland of *L. camtschaticum*. Among them, 4306 endogenous peptides had fewer than 10 amino acids, 179 had 10–15 amino acids, and 43 had more than 15 amino acids (Table 1). For a complete list of endogenous peptides and proteins, please refer to Appendix A.

Subcellular Location Analysis showed that most precursor proteins were located in nuclear (497), cytoplasmic (239), plasma membrane (235), extracellular (128), and mitochondria (108) (Figure 1A). Precursor proteins were classified using a Cluster of Orthologous Groups (COG) of proteins. According to the COG analysis, the precursor proteins were mostly involved in signal transduction, general function prediction, post-translational modifications, protein turnover, chaperoning, and transcription (Figure 1B). To predict the potential functions of the endogenous peptides, we performed Gene Ontology (GO) and Kyoto Encyclopedia of Genes and Genomes (KEGG) pathway analyses of their precursor proteins. As shown in Figure 2, the analysis of biological processes revealed that these precursor proteins were mainly involved in developmental processes, cellular component organization, and anatomical structure development, and the analysis of cellular components showed that these proteins were mainly distributed in the cytoskeleton and supramolecular complex. In addition, molecular function analysis showed that these proteins were mainly related to cell adhesion molecule binding. The precursor proteins were significantly enriched in the KEGG pathways of the PI3K-Akt signaling pathway (map04151), regulation of the actin cytoskeleton (map04810), and focal adhesion (map04510) (Figure 3).

### 2.2. In Silico Identification and Synthesis of Candidate Bioactive Functional Peptides

To identify potential bioactive functional peptides, 4528 endogenous peptides from the buccal glands of lamprey were screened using bioactive peptide prediction websites (Table 2). Four candidates predicted bioactive peptides, including two anti-inflammatory peptides, VSLNLPYSVVRGEQFVVQA (Figure 4A) and DIPVPEVPILE (Figure 4B), one antibacterial peptide VVQLPPVVLGTFG (Figure 4C), and an ACE inhibitor, VPPPPLVLPPASVK (Figure 4D), were synthesized and examined for purity using high-performance liquid chromatography (HPLC). The chromatograms of the synthesized peptides (purity ≥ 90%) are shown in Appendix A, and the MS/MS spectra of the synthesized peptides are shown in Appendix A.

### 2.3. Validation of the Bioactivity of Potential Anti-Inflammatory Peptides

To investigate whether VSLNLPYSVVRGEQFVVQA and DIPVPEVPILE possess anti-inflammatory properties, the expression of *TGF-β*, *IL-10*, *TNF-α*, *IL-1β*, and *MCP-1* in THP-1 cells was detected using qPCR. As shown in Figure 5A, the expression of the anti-inflammatory factors *TGF-β* and *IL-10* decreased significantly after lipopolysaccharide (LPS) stimulation, while that of *TNF-α*, *IL-1β*, and *MCP-1* increased significantly. However, after incubation with VSLNLPYSVVRGEQFVVQA for 1, 3, and 6 h, the expression of *TGF-β* and *IL-10* in THP-1 cells increased significantly, whereas that of *TNF-α*, *IL-1β*, and *MCP-1* decreased significantly compared to that of the group stimulated with only LPS. This indicated that the VSLNLPYSVVRGEQFVVQA peptide possesses significant anti-inflammatory activity. *TGF-β*, *IL-10*, *TNF-α*, *IL-1β*, and *MCP-1* were all significantly upregulated in THP-1 after incubation with DIPVPEVPILE for 1, 3, and 6 h compared to that in the LPS-stimulated group (Figure 5B). This indicated that the DIPVPEVPILE peptide did not possess effective anti-inflammatory activity. To further verify the anti-inflammatory effect of VSLNLPYSVVRGEQFVVQA, the expression of TNF-α and TGF-β was detected using western blotting. As shown in Figure 5C, compared to the group treated with LPS only, the expression of TNF-α protein was significantly downregulated in THP-1 cells incubated with VSLNLPYSVVRGEQFVVQA, and the expression of TGF-β protein was significantly upregulated in THP-1 cells incubated with VSLNLPYSVVRGEQFVVQA for 3 h.

### 2.4. Validation of the Bioactivity of the Potential Antimicrobial Peptide

*Escherichia coli* and *Staphylococcus aureus* were incubated with 500 μM (final concentration) of the candidate antimicrobial peptide, VVQLPPVVLGTFG, to test its antimicrobial activity. As shown in Figure 6A,B, different concentrations of VVQLPPVVLGTFG inhibited *S. aureus* to a certain extent but did not affect *E. coli*. The results of confocal laser-scanning microscopy (Figure 6C) verified the killing effect of the peptide VVQLPPVVLGTFG on *S. aureus.*

### 2.5. Validation of the Bioactivity of the Potential ACE Inhibitory Peptide

The activity of the candidate ACE inhibitor, VPPPPLVLPPASVK, was determined using a human ACEI inhibitor ELISA kit and compared with that of the positive control. Appendix A shows the standard curve for the VPPPPLVLPPASVK concentrations and optical density values. As shown in Figure 7, VPPPPLVLPPASVK showed inhibitory activity similar to that of the positive control at low concentrations, although correlation with dose was not observed.

## 3. Discussion

To feed successfully, blood-sucking animals must bypass or suppress the host’s defense mechanisms, particularly the immune and coagulation systems. Typical hematophagous animals, including leeches and ticks, contain biologically active compounds in their secretions, especially in the saliva. The components of the salivary glands of leeches and ticks are relatively well studied, and classical active peptides, such as hirudin, have been used widely [25,26]. As bloodsuckers, the buccal glands of lampreys are considered a source of diverse bioactive proteins, from which various anticoagulant- and immune-related proteins, such as RGD (Arg-Gly-Asp) motif toxin and cysteine-rich secretory protein (CRISP), have been identified in recent years [14,27]. However, studies on endogenous peptides present in the buccal gland of lampreys are limited. The buccal glands of lampreys are a good source of endogenous peptides because of their large size and ease of accessing buccal gland secretions. Previous studies have shown that up to 50 µL secretion can be obtained from the buccal glands of one *L. camtschaticum* and that the measured protein concentration can exceed 100 µg/µL [28]. This is an advantage for peptidomic analysis, as false-negative results for low-abundance peptides can be avoided, and enough peptides can still be recovered after multiple ultrafiltration and desalting. The results of a peptidomic analysis showed that the endogenous peptides were abundant (4528 in number) in the buccal gland of the lamprey, much more compared to other peptidomic publications, which is consistent with the high protein concentration in the buccal gland and also possibly due to the high physiological activity in the buccal gland during blood sucking [27]. However, most of the identified peptides (4306) had fewer than 10 amino acids and were not abundant, indicating that they might have been generated via natural protein degradation. Therefore, we focused on selecting peptides with >10 amino acids and high expression levels for validating the function.

Bioinformatic analysis of buccal gland precursor proteins helps us to understand the origin of endogenous peptides and to speculate on their potential functions. GO and COG analyses showed a functional abundance of precursor proteins in the buccal glands, which is consistent with the large number of precursor proteins (1224). The precursor protein-enriched PI3K-Akt signaling pathway and the regulatory actin cytoskeleton pathway are involved in a variety of physiological functions, including cell proliferation, angiogenesis, and metabolism. Activation of the PI3K-Akt signaling pathway has been reported to promote blood circulation [29]. Some of the peptides we detected are derived from key proteins of the regulatory actin cytoskeleton pathway, such as Rho GTPase, which has been reported to regulate platelet aggregation and spreading [30]. In addition, the focal adhesion pathway is associated with the immune response in fish [31]. These signaling pathways are closely related to the blood-sucking properties of the lamprey and the regulation of cell proliferation, and angiogenesis also makes these pathways relevant to the development and progression of cancer, suggesting that the anticancer potential of endogenous peptides in the buccal gland of lamprey deserves investigation in the future.

Currently, large-scale analysis of omics data is considered a promising strategy for identifying bioactive peptides, for example, the screening for antimicrobial peptides in the transcriptome and genome [32,33,34]. For blood-sucking leeches, researchers have used computational algorithms in medicinal leech genome assembly to identify amino acid sequences encoding potential antimicrobial peptides [32]. The use of in silico methods for omics data to predict bioactive peptides effectively reduces the time and cost of experimental analysis, especially when the raw data are large. The increase in the number of websites currently available for analyzing peptide bioactivity poses problems to users. Several websites are usually available for determining biological activity, and the activity predictions from these sites are often inconsistent owing to differences in algorithms and database size. A peptide may be predicted to have high activity at one site and low activity at another, which may confound the user. Hence, ten online websites were used to predict the anti-inflammatory, antibacterial, and ACE inhibitory activities in this study. Each synthesized candidate peptide was predicted to be active by at least three websites. In short, the current peptide activity screening websites are inconvenient for users with limited computer skills. In addition, bioactive peptides screened using prediction websites are not always accurate. The screened anti-inflammatory peptide (DIPVPEVPILE) was predicted to be active by several prediction websites but was completely ineffective in our anti-inflammatory assays.

The anti-inflammatory effects of the leech and tick salivary gland secretions have been described previously [35,36]. Both hirudin and Iripin-1 (Ixodes ricinus serpin-1) have been reported to possess anti-inflammatory functions [36,37]. Notably, the serpin gene is also present in lampreys but is mainly expressed in leukocytes and the liver [38]. The candidate anti-inflammatory peptide, VSLNLPYSVVRGEQFVVQA, significantly decreased the expression of pro-inflammatory factors and elevated the expression of anti-inflammatory factors. The precursor protein of this peptide is CD109, which has been shown to have anti-inflammatory properties [39]. However, whether the anti-inflammatory mechanism is similar to that of CD109 remains unclear. Since CD109 has been reported as a novel TGF-β co-receptor and TGF-β antagonist, the mechanism of TGF-β upregulation by VSLNLPYSVVRGEQFVVQA may be more complicated [40,41]. Some proteins with antibacterial activity, such as destabilase-lysozyme (mlDL), have been found in the salivary gland secretions of leeches. However, antimicrobial peptides from salivary glands have rarely been reported [42]. The candidate antimicrobial peptide screened in this study inhibited the growth of *S. aureus* but was not broad-spectrum in nature. The peptide VVQLPPVVLGTFG has neither aromatic amino acids nor cysteine, and there are very few reports related to the function of its precursor protein, carnosine dipeptidase 2. Therefore, its antibacterial mechanism is not clear. In the future, we plan to evaluate the modification potential of this peptide and continue to screen for antimicrobial peptides in lamprey. The candidate ACE inhibitory peptide, VPPPPLVLPPASVK, showed better activity at low concentrations than the positive control but surprisingly did not show a dose correlation. This may be due to the high concentration of the peptide used. The ELISA-based conclusions are still relatively preliminary, and owing to time and experimental costs, only a primary validation of the activity was performed. We plan to validate this peptide using additional experiments.

In summary, we analyzed endogenous peptides from the buccal glands of *L. camtschaticum* using peptidomics analysis and predicted bioactive peptides using websites. Validation of the four peptides screened showed that the endogenous peptides in the buccal glands of lamprey possessed various biological activities. Our study revealed that the buccal glands of lampreys are a potential source of multiple bioactive peptides; furthermore, it provides new insights regarding the blood-sucking mechanisms of lamprey.

## 4. Materials and Methods

### 4.1. Experimental Animals and Sample Collection

The adult *Lethenteron camtschaticum* (body length: 35–45 cm) used in this study were caught in Jilin Province, China, and then raised in a laboratory tank equipped with a physical and biological filtration system set to 4 °C. Extraction of secretion from the buccal glands of lamprey with the syringe.

### 4.2. Peptide Extraction

Peptide extraction was carried out using the standard operating procedure developed by Biotree Biotech. Secretion from the buccal gland was added grinding beads (5.2 mm JX-GZ0138, Jingxin Technology, Shanghai, China) and ground for 4 min, sonicated in an ice water bath for 5 min, repeated thrice. Immediately afterward, three times the volume of pre-cooled methanol was added and vortexed to precipitate the proteins. It was centrifuged at 12,000 rpm, 4 °C for 10 min, and the supernatant was transferred to a new tube. Subsequently, the supernatant solution was aspirated to 500 µL and ultrafiltered to 100 µL using a 3 kDa ultrafiltration tube (Ultra-4 Centrifugal Filter Unit-3 KDa, Merck, Darmstadt, Germany) at 10,000 rpm. The filtrate was collected repeatedly, and then a C18 solid phase extraction cartridge (SLBC181100, Biocomma Technology, Shenzhen, China) was used to desalt the sample. The cartridge was activated by adding 500 μL buffer C, allowing the entire solution to flow slowly into the centrifuge tube. Then, 500 μL buffer A was added to equilibrate, and the sample was added. After adding 500 μL buffer A for 2 washes, 200 μL buffer B was added for elution. Samples were reduced with 5 mM dithiothreitol for 25 min at 37 °C and subsequently alkylated with iodoacetamide for 25 min at 37 °C. Then, it was centrifuged at 12,000 rpm, 4 °C for 10 min, and the supernatant was collected and dried overnight at 4 °C under a vacuum. The experiment contains three biological replicates. The overall experimental design of this study is shown in Figure 8.

### 4.3. NanoLC-MS/MS Analysis and Database Search

For each sample, 2 μL of total peptides were separated and analyzed with a nano UPLC (EASY-nLC1200, Thermo Fisher Scientific, Waltham, MA, USA) coupled to a Q Exactive HFX Orbitrap instrument (Thermo Fisher Scientific, Waltham, MA, USA) with a nanoelectrospray ion source. Separation was performed using a reversed-phase column (100 μm ID ×15 cm, Reprosil-Pur 120 C18 AQ, 1.9 μm, Dr. Maisch, Tubingeng, Germany). Mobile phases were H_2_O with 0.1% FA, 2% ACN (phase A), and 80% ACN, 0.1% FA (phase B). Separation of the sample was executed with a 60 min gradient at 300 nL/min flow rate. Gradient B: 2–5% for 2 min, 5–22% for 34 min, 22–45% for 20 min, 45–95% for 2 min, 95% for 2 min. Data-dependent acquisition (DDA) was performed in profile and positive mode with Orbitrap analyzer at a resolution of 120,000 and m/z range of 350 to 1600 for MS1; for MS2, the resolution was set to 15,000 with a dynamic first mass. The top 10 most intense ions were fragmented by HCD, with normalized collision energy (NCE) of 30% and an isolation window of 1.2 m/z. The dynamic exclusion time window was 30 s; single-charged peaks and peaks with a charge exceeding 6 were excluded from the DDA procedure. The vendor’s raw MS files were processed using Proteome Discoverer (PD) software (Version 2.4.0.305, Thermo Fisher Scientific, Waltham, MA, USA) and the built-in Sequest HT search engine. MS spectra lists were searched against their species-level UniProt FASTA databases (UniProt-Petromyzon marinus_10375.fasta), Carbamidomethyl [C] as a fixed modification, Oxidation (M) and Acetyl (Protein Nterm) as variable modifications. The false discovery rate (FDR) was set to 0.01 for both PSM and peptide levels. Peptide identification was performed with an initial precursor mass deviation of up to 10 ppm and a fragment mass deviation of 0.02 Da.

### 4.4. Bioinformatics Analysis

The peptide isoelectric point (PI) and molecular weight (MW) information were obtained online (https://web.expasy.org/protparam/, accessed on 25 April 2023). Subcellular Location Analysis was carried out by PSORTb (v3.0, Brinkman Laboratory, Simon Fraser University, British Columbia, Canada). The Proteome Annotated Gene Selection (GO) was obtained from the UniProt GOA database (https://www.ebi.ac.uk/GOA/, accessed on 6 July 2022). Information was analyzed from UniProtKB/SwissProt, Kyoto Encyclopedia of Genes and Genomes (KEGG), and Gene Ithology (GO) to analyze the functional enrichment of the identified precursor proteins of the endogenous peptides. HeliQuest was used to analyze the spiral wheel projections.

### 4.5. Bioactive Peptide Prediction

Three anti-inflammatory peptide online prediction tools, AntiInflam [43], AIPpred [44], and PreAIP [45], were used to perform a comprehensive analysis of peptides (amino acid ≥ 10). CAMPR4 [46], Antimicrobial Peptide Scanner vr.2 [47], and AmpGram [48] were used to predict antimicrobial peptides. ACE inhibitory peptide screening was performed based on the method of Ezequiel et al. [49]. Peptides were first ranked by PeptideRanker (http://distilldeep.ucd.ie/PeptideRanker/, accessed on 5 November 2022) and then analyzed using BIOPEP-UWM [50], athtpin, and mAHTPred [51]. The online website for predicting peptide bioactivity and the selection standard of bioactive peptides is shown in Table 2.

### 4.6. Peptides Synthesis

Peptides used in this study were synthesized via N-9-fluorenylmethyloxycarbonyl (Fmoc) chemistry using an AAPPTec peptide synthesizer (Titan 357) by Nanjing Synpeptide Co., Ltd., Nanjing, China, and purified using AKTA pure chromatography system (GE Healthcare, Chicago, IL, USA). The purity of peptides was analyzed by Waters reversed-phase high-performance liquid chromatography system (Waters e2695, Milford, CT, USA). The synthesized peptides were diluted in sterilized ultrapure water and stored at −20 °C. 

### 4.7. Cell Culture and Treatment

Human monocyte THP-1 cells were purchased from Fenghui Biotechnology Co., Ltd., Hunan, China. THP-1 cells were cultured in Roswell Park Memorial Institute-1640 media (RPMI-1640) containing 10% (*v*/*v*) fetal bovine serum (FBS) and 1% penicillin-streptomycin (P/S) at 37 °C with 5% CO_2_. THP-1 cells were incubated with 100 μM screened anti-inflammatory peptides (VSLNLPYSVVRGEQFVVQA, DIPVPEVPILE) at 37 °C in 5% CO_2_ for 1 h, 3 h, and 6 h. Lipopolysaccharide (LPS, 100 ng/mL) was used as the positive control.

### 4.8. Real-Time Quantitative PCR

Tumor Necrosis Factor-alpha (TNF-α), Interleukin-1β (IL-1β), Transforming growth factor-β (TGF-β), Interleukin-10 (IL-10), and Monocyte Chemoattractant Protein-1 (MCP-1) gene were selected for QPCR. Primer Premier 5.0 was used to create primers for real-time quantitative PCR analysis (Table 3). The efficiency of primers varied from 90% to 105%. QPCR was carried out in 20 μL reactions of 2× chamQ SYBR Color qPCR Master Mix (low ROX Rremixed) (Vazyme Biotech, Nanjing, China), forward primer (10 μM), reverse primer (10 μM), and cDNA 200 ng. qTOWER 2.0 Real-Time PCR System (Thermo Fisher Scientific, Waltham, MA, USA) was applied to the real-time quantitative PCR. The following program was used for real-time quantitative PCR: 30 s at 95 °C, 40 cycles of 10 s at 95 °C, and 30 s at 60 °C. The difference in threshold cycle (ΔCT) was calculated by subtracting the average CT of β-actin mRNA from the target gene’s average CT.

### 4.9. Western Blotting

Cells were collected with centrifugation at 1000 rpm and then lysed in RIPA lysis buffer. An equal amount of cell lysates was added to the SDS-PAGE gel and transferred to the PVDF membrane. The membrane was detected with primary antibodies against TNF-α, TGF-β, and β-actin at 1:2000 dilution, followed by a 1:5000 dilution of secondary antibody. Protein bands were detected by ECL. The antibodies used in this study were purchased from Proteintech, Wuhan, China.

### 4.10. Minimal Inhibitory Concentrations (MIC)

An amount of 50 μL of the diluted antimicrobial peptide was added on a 96-well plate in a concentration gradient to a final concentration from 1.57 μM to 100 μM. Then, 50 μL of the diluted bacterial solution was added to a final concentration of 1 × 10^5^ CFU/mL. After incubation at 37 °C for 12 h, we determined the MIC of the peptide by measuring the absorbance at 600 nm with an enzyme marker. The strains used in this study were *Staphylococcus aureus* and *Escherichia coli*.

### 4.11. Confocal Laser-Scanning Microscopy

*E. coli* and *S. aureus* were incubated with a final concentration of 500 μM peptide at 37 °C for 1 h. Then, 3.5 μM SYTO 9 and propidium iodide (PI; 1:1000) were added to the treated bacterial suspensions and incubated for 30 min at room temperature, protected from light. Stained bacteria were visualized using a confocal laser-scanning microscope (Carl Zeiss AG, Oberkochen, Germany).

### 4.12. ACE Inhibitory Activity

The activity of the ACE inhibitory peptide was measured using the Human Angiotensin-Converting Enzyme Inhibitor (ACEI) ELISA kit (Fantai Biotechnology, Shanghai, China) according to the manufacturer’s instructions. The concentration gradients of ACE inhibitory peptide were set at 5 μM, 10 μM, 25 μM, 50 μM, and 100 μM, respectively.

### 4.13. Statistical Analysis

Data were analyzed using GraphPad Prism 9 (graphpad prism, San Diego, CA, USA), and all results were expressed as the mean of three independent experiments, using three copies of each experiment. The values shown were expressed as the mean ± SD based on independent experiments. The significance of the difference between the two was assessed for each group using the SPSS statistical package (IBM, Armonk, NY, USA). This difference was statistically significant or highly significant, ns *p* > 0.05, * *p* < 0.05, ** *p* < 0.01, *** *p* < 0.001, **** *p* < 0.0001, respectively.

## Figures and Tables

**Figure 1 marinedrugs-21-00389-f001:**
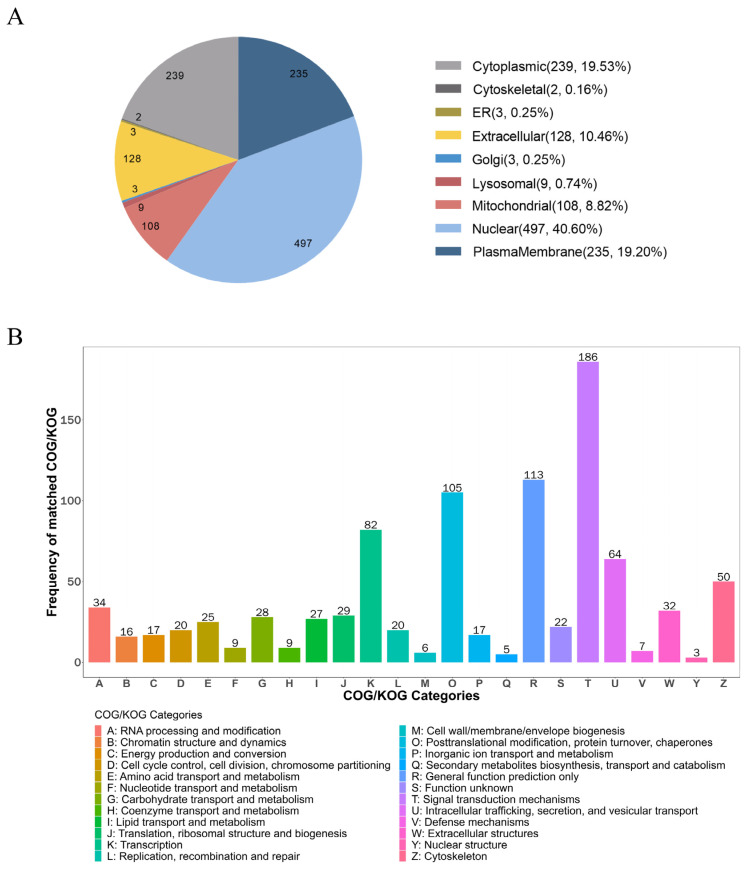
Subcellular localization and COG analysis of endogenous peptide precursor proteins. (**A**) Distribution of subcellular structures of endogenous peptide precursor proteins. The pie chart shows the number and proportion of mapped peptide precursor proteins in each subcellular compartment. (**B**) COG analysis of peptide precursor proteins. The horizontal axis represents the different classification contents of COG, while the vertical axis represents the frequency of COG.

**Figure 2 marinedrugs-21-00389-f002:**
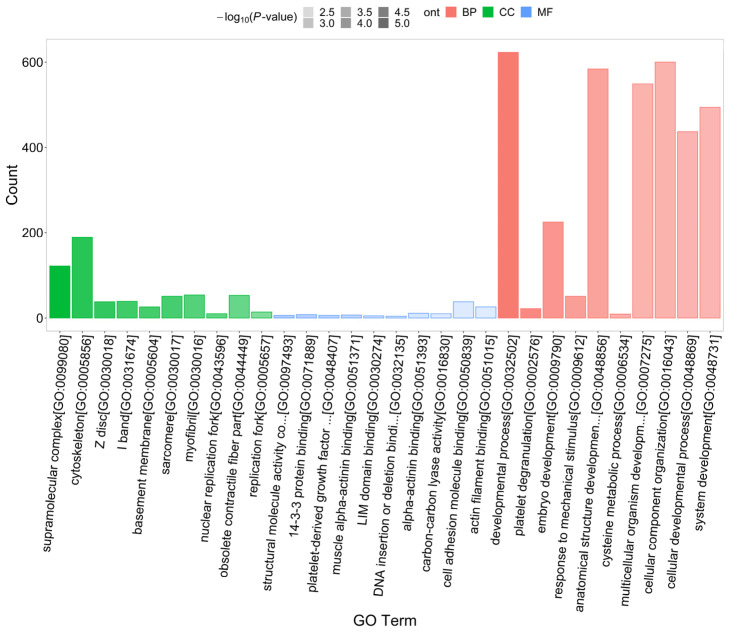
GO analysis of endogenous peptide precursor proteins. The horizontal coordinate is the GO term, and the vertical coordinate is the number of mapped peptide precursor proteins. Red represents information regarding biological process annotation, green represents information on cellular composition annotation, blue represents information on molecular function annotation, and transparency represents *p*-value size. The darker the color, the smaller the *p*-value.

**Figure 3 marinedrugs-21-00389-f003:**
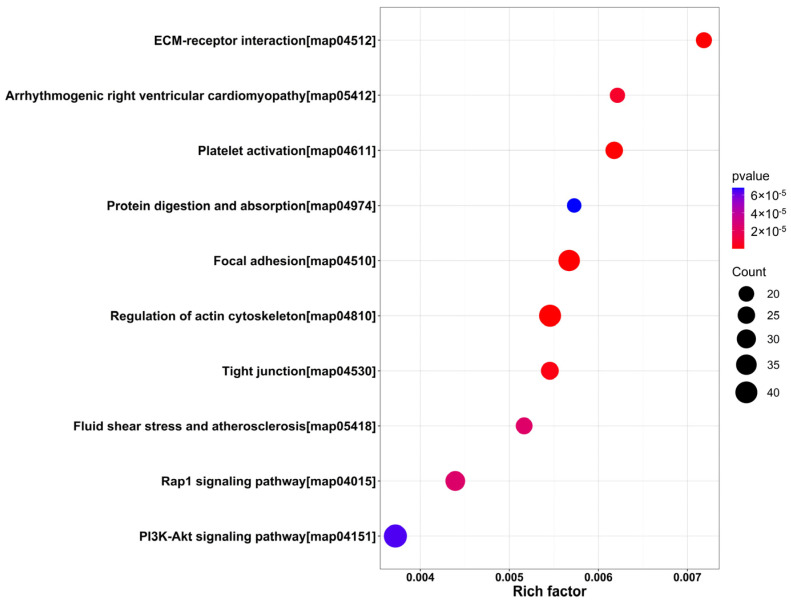
KEGG analysis of endogenous peptide precursor proteins. The horizontal axis represents the enrichment degree Rich factor value, and the vertical axis represents the KEGG pathway information. Among them, the size of the circle represents the number of peptide precursor proteins in the pathway; the larger the circle, the more the number; the color of the circle represents the corrected *p*-value size; for example, the redder the color, the smaller the *p*-value.

**Figure 4 marinedrugs-21-00389-f004:**
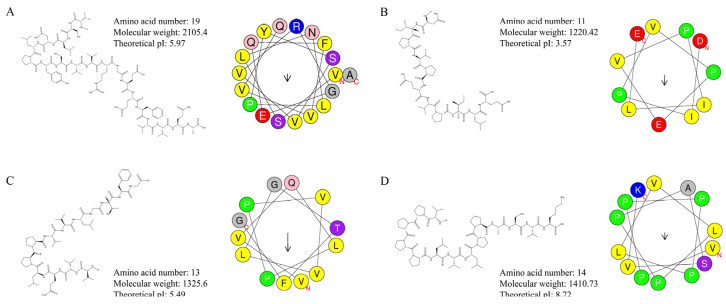
Peptide information of VSLNLPYSVVRGEQFVVQA, DIPVPEVPILE, VVQLPPVVLGTFG, and VPPPPLVLPPASVK. (**A**) Candidate anti-inflammatory peptide VSLNLPYSVVRGEQFVVQA. (**B**) Candidate anti-inflammatory peptide DIPVPEVPILE. (**C**) Candidate antibacterial peptide VVQLPPVVLGTFG. (**D**) Candidate ACE inhibitor VPPPPLVLPPASVK. Yellow represents hydrophobic residues, purple represents serine and threonine residues, dark blue represents basic residues, red represents acidic residues, pink represents asparagine and glutamine residues, gray represents alanine and glycine residues, light blue represents histidine residues, and green represents proline residues. The arrows indicate the direction of the hydrophobic moment (µH).

**Figure 5 marinedrugs-21-00389-f005:**
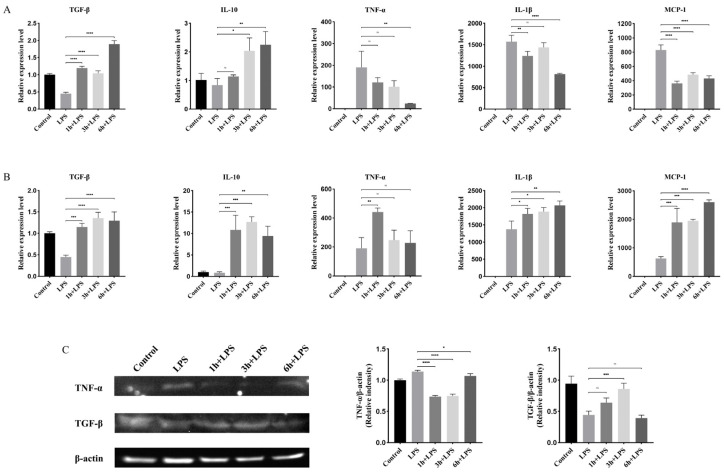
Functional verification of VSLNLPYSVVRGEQFVVQA and DIPVPEVPILE peptides. (**A**) Effect of VSLNLPYSVVRGEQFVVQA peptide on the expression of LPS-stimulated THP-1 cytokines. (**B**) Effect of DIPVPEVPILE peptide on the expression of LPS-stimulated THP-1 cytokines. ^ns^
*p* > 0.05, * *p* < 0.05, ** *p* < 0.01, *** *p* < 0.001, **** *p* < 0.0001. (**C**) Expression of TNF-α and TGF-β in THP-1 cells after incubation with VSLNLPYSVVRGEQFVVQA.

**Figure 6 marinedrugs-21-00389-f006:**
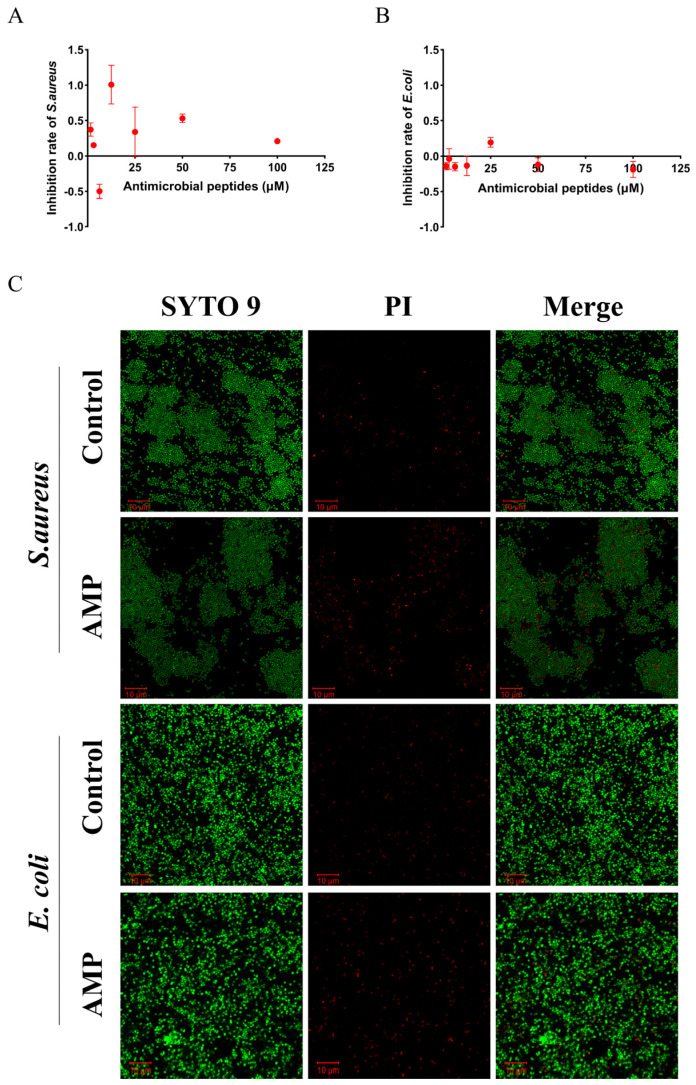
Antibacterial activity of VVQLPPVVLGTFG peptide. (**A**) Antibacterial activity of VVQLPPVVLGTFG against *S. aureus*. (**B**) Antibacterial activity of VVQLPPVVLGTFG against *E. coli*. (**C**) Confocal laser-scanning microscopy of *S. aureus* and *E. coli* after VVQLPPVVLGTFG treatment. Green represents live bacteria, and red represents dead bacteria.

**Figure 7 marinedrugs-21-00389-f007:**
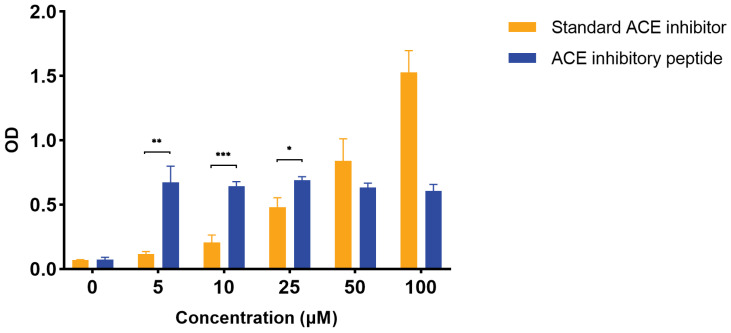
ACE inhibitory activity of the VPPPPLVLPPASVK peptide. Different concentrations (5 μM, 10 μM, 25 μM, 50 μM, and 100 μM) of VPPPPLVLPPASVK peptide were set to detect ACE inhibitory activity. * *p* < 0.05, ** *p* < 0.01, *** *p* < 0.001.

**Figure 8 marinedrugs-21-00389-f008:**
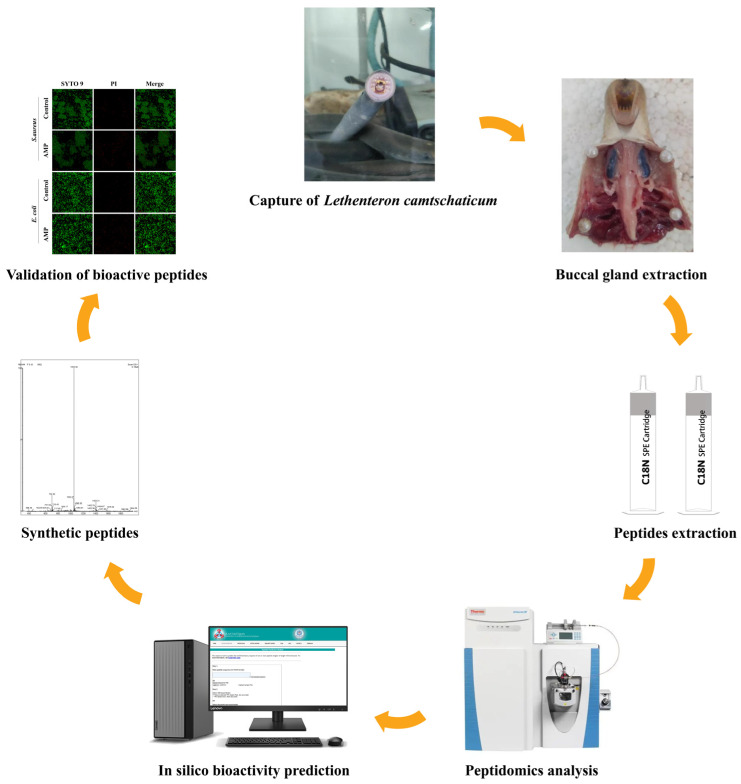
Process of screening and verifying the function of bioactive peptides in the buccal gland of lamprey.

**Table 1 marinedrugs-21-00389-t001:** Overall endogenous peptides.

Peptides	Precursor Protein	<10 aa	10–15 aa	>15 aa
4528	1224	4306	179	43

**Table 2 marinedrugs-21-00389-t002:** Bioactive peptide prediction online website.

Bioactivity Prediction	Website	Accesse Date	Selection Standard
Anti-inflammatory peptide prediction:			
AntiInflam	http://metagenomics.iiserb.ac.in/antiinflam/	26 October 2022	>0.5
AIPpred	http://www.thegleelab.org/AIPpred/	27 October 2022	>0.36
PreAIP	http://kurata14.bio.kyutech.ac.jp/PreAIP/	27 October 2022	Score ≥ 0.468 High Confidence 0.468 > Score ≥ 0.388 Medium Confidence 0.388 > Score ≤ 0.342 Low Confidence
Antimicrobial peptide prediction:			
CAMPR4	http://camp.bicnirrh.res.in/	23 October 2022	≥0.5
Antimicrobial Peptide Scanner vr.2	https://dveltri.com/ascan/v2/ascan.html	22 October 2022	>0.5
AmpGram	http://biongram.biotech.uni.wroc.pl/AmpGram/	24 October 2022	>0.5
ACE inhibitory peptide prediction:			
BIOPEP-UWM	https://biochemia.uwm.edu.pl/biopep-uwm/	6 November 2022	ACE inhibition positive
AHTpin	https://webs.iiitd.edu.in/raghava/ahtpin/	7 November 2022	>0.0
mAHTPred	http://thegleelab.org/mAHTPred	7 November 2022	>0.44

**Table 3 marinedrugs-21-00389-t003:** List of qPCR primers used in this study.

Gene Name	Forward Primers (5′-3′)	Reverse Primers (5′-3′)
*TNF-α*	AAGGACACCATGAGCACTGAAAGC	AGGAAGGAGAAGAGGCTGAGGAAC
*IL-1β*	GGACAGGATATGGAGCAACAAGTGG	TCATCTTTCAACACGCAGGACAGG
*TGF-β*	TATTGAGCACCTTGGGCACTGTTG	CCTTAACCTCTCTGGGCTTGTTTCC
*IL-10*	GCCGTGGAGCAGGTGAAGAATG	ATAGAGTCGCCACCCTGATGTCTC
*MCP-1*	GGCTGAGACTAACCCAGAAACATCC	GGGAATGAAGGTGGCTGCTATGAG
*β-actin*	CAGATGTGGATCAGCAAGCAGGAG	CGCAACTAAGTCATAGTCCGCCTAG

## Data Availability

The original data presented in this study are included in the article/Appendix A; further inquiries can be directed to the corresponding author.

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
