# Peer review of "Peptidomics Analysis Reveals the Buccal Gland of Jawless Vertebrate Lamprey as a Source of Multiple Bioactive Peptides"

_marinedrugs, 2023, doi:10.3390/md21070389_

Round 1

Reviewer 1 Report

The article presented by Wang and colleagues performed a peptidomic analysis of the oral gland of lampreys, identifying thousands of peptides, precursor protein fragments residing in different subcellular compartments and involved in various biological processes, according to the results of the analyses performed in GO and KEGG. Furthermore, after predicting biological activity based on structural information, 4 peptides were synthesized and tested for anti-inflammatory, antibacterial, and angiotensin-converting enzyme inhibitory activity. The article is interesting, however, it has some deficiencies in terms of result presentation and discussion, as well as loss of result information and methodological description. Several points need to be improved.

  1. The abstract is lacking in terms of presenting the obtained results, both in descriptive terms of peptidomics and bioactivity testing. In this section, a brief summary should be provided, including quantitative data characterizing the anti-inflammatory or antibacterial action, as well as the corresponding peptide sequence that exhibited this activity.

  2. Topics 2.1 and 2.2 could be presented as a single topic. However, it is important to note that the supplementary Table S1, which is crucial for the study, was not attached, making it impossible to verify the quality of the mass spectrometry data and peptide sequencing. The results from Figures 1, 2, and 3 are interesting and well-presented, but they lack thorough discussion. The authors did not provide any correlation between these data and the functionality of the gland, nor did they compare their findings with existing literature on the content of this type of secretion. Another point to consider is that the peptidomic analysis was performed in triplicate, but quantitative data among the samples were not presented. Furthermore, quantitative and qualitative information regarding post-translational modifications and the position of peptides within precursor proteins (N-terminal, internal, and C-terminal) was also not provided.

  3. Despite the authors using different in silico tools for predicting bioactivity, they also failed to discuss the structural aspects that led to the selection of the 4 peptides. In addition to HPLC, the MS/MS spectra of the 4 synthetic peptides should be included in the supplemental material.

  4. Regarding section 4.2 on the peptide extraction methodology, it appears to be incomplete. Typically, peptidomic analyses involve a step to inhibit post-mortem degradation, either through chemical or physical protease inhibition methods, followed by the use of molecular weight cutoff filters below 10 kDa, and then desalting and peptide concentration using C18 columns. In line 267, the sentence "A C18 column (molecular masses below 3KDa)" seems to be incorrect.

  5. Were the peptide extracts reduced with DTT and alkylated with iodoacetamide for the fixed modification Carbamidomethyl [C]? Based on the information provided, if the methodology did not explicitly mention the reduction of peptide extracts with DTT and alkylation with iodoacetamide for the fixed modification Carbamidomethyl [C], it can be considered a gap in the methodology description. Generally, these steps are standard in sample preparation for proteomic/peptidomic analysis, especially to preserve cysteine residues and prevent their oxidation during processing. 

  6. Regarding the results of antimicrobial activity, it is not clear what the effective minimum inhibitory concentration was. Additionally, why was only a concentration of 500 μM used for the confocal microscopy experiment? In the image, it is not possible to visualize the staining with propidium iodide (PI).

  7. Figure 7 does not present any statistical data, although it is indicated in the methodology that the experiment was performed in triplicate.

  8. The discussion is too superficial regarding the bioactivity results.

Author Response

Reviewer 1

Question 1: 

The abstract is lacking in terms of presenting the obtained results, both in descriptive terms of peptidomics and bioactivity testing. In this section, a brief summary should be provided, including quantitative data characterizing the anti-inflammatory or antibacterial action, as well as the corresponding peptide sequence that exhibited this activity.

Response 1:

Thank you for your suggestion. We have reworded our abstract. However, the journal's author guidelines do not support a brief summary, and we can only attempt to provide as much information as possible in the abstract. Please check our revised manuscript.

Question 2: 

Topics 2.1 and 2.2 could be presented as a single topic. However, it is important to note that the supplementary Table S1, which is crucial for the study, was not attached, making it impossible to verify the quality of the mass spectrometry data and peptide sequencing. The results from Figures 1, 2, and 3 are interesting and well-presented, but they lack thorough discussion. The authors did not provide any correlation between these data and the functionality of the gland, nor did they compare their findings with existing literature on the content of this type of secretion. Another point to consider is that the peptidomic analysis was performed in triplicate, but quantitative data among the samples were not presented. Furthermore, quantitative and qualitative information regarding post-translational modifications and the position of peptides within precursor proteins (N-terminal, internal, and C-terminal) was also not provided.

Response 2:

Thank you for your suggestion, we have combined 2.1 and 2.2 into a single section. We did upload supplementary Table S1 in our supplementary material, but it was not in the supplemental materials received by the reviewer. We have again submitted Table S1 in the revised manuscript, including the peptide sequencing, the position of peptides within precursor proteins and quantitative information of the peptidomics, so if you still do not see Table S1 in the revised manuscript, please contact the editor. We will also remind the editor to ensure that the reviewers receive Table S1.

We have added the relevant information to Figures 1, 2, and 3 in the discussion section.

Question 3: 

Despite the authors using different in silico tools for predicting bioactivity, they also failed to discuss the structural aspects that led to the selection of the 4 peptides. In addition to HPLC, the MS/MS spectra of the 4 synthetic peptides should be included in the supplemental material.

Response 3:

Thank you for your suggestion. We have added the selection standards for predicting bioactivity in Table 2. We also added the MS/MS spectra of the 4 synthetic peptides in the supplemental material as Figure S2 in our revised manuscript. Please check our revised manuscript.

Question 4: 

Regarding section 4.2 on the peptide extraction methodology, it appears to be incomplete. Typically, peptidomic analyses involve a step to inhibit post-mortem degradation, either through chemical or physical protease inhibition methods, followed by the use of molecular weight cutoff filters below 10 kDa, and then desalting and peptide concentration using C18 columns. In line 267, the sentence "A C18 column (molecular masses below 3KDa)" seems to be incorrect.

Response 4:

We have rewritten section 4.2 to improve the peptide extraction method and correct some mistakes. Please check our revised manuscript.

Question 5: 

Were the peptide extracts reduced with DTT and alkylated with iodoacetamide for the fixed modification Carbamidomethyl [C]? Based on the information provided, if the methodology did not explicitly mention the reduction of peptide extracts with DTT and alkylation with iodoacetamide for the fixed modification Carbamidomethyl [C], it can be considered a gap in the methodology description. Generally, these steps are standard in sample preparation for proteomic/peptidomic analysis, especially to preserve cysteine residues and prevent their oxidation during processing.

Response 5:

Thank you for your question. We previously ignored this in the methods section. We have revised the method in the revision. Please check section 4.2 of our revised manuscript.

Question 6: 

Regarding the results of antimicrobial activity, it is not clear what the effective minimum inhibitory concentration was. Additionally, why was only a concentration of 500 μM used for the confocal microscopy experiment? In the image, it is not possible to visualize the staining with propidium iodide (PI).

Response 6:

Thank you for your question. The aim of this study was to reveal the medicinal potential of endogenous peptides in the buccal glands of lamprey. We prefer to express the diversity of biological activities of these peptides than to study in depth a particular function. The antimicrobial experiments in this paper were indeed relatively simple, as the results showed that the antimicrobial peptides we selected, although killing S. aureus, did not have an advantageous effect compared to the antimicrobial peptides of other species. Due to time and funding limitations, we did not study them in depth. So we chose only a concentration of 500 μM for the confocal microscopy experiment just to demonstrate the antibacterial activity of our synthesized peptide. We hope to perform more comprehensive and in-depth studies on the antimicrobial peptides of lamprey in the future.

Regarding PI staining, this is a technique that has emerged in the last few years and can be used in conjunction with Annexin-V to differentiate between cells in the early and late stages of apoptosis and dead cells (bacteria). PI cannot enter normal cell membranes but can enter incomplete cell membranes and stain the nucleus, while SYTO9 can stain all bacteria. Therefore, PI is mainly used in antibacterial experiments to stain bacteria with incomplete cell membranes (dead bacteria). When SYTO9 and PI are used together, live bacteria can only be stained by SYTO9 (green fluorescence), while dead bacteria can be stained by SYTO9 and PI, and the fluorescence intensity of PI is stronger than SYTO9, so dead bacteria show red fluorescence.

Our Figure 6 may have been smaller and the red fluorescence of PI was not significant, we have adjusted the size of Figure 6 and now the red fluorescence of PI is more visible. We have also provided several publications for your reference on antimicrobial assays using SYTO9 and PI staining.

Shen L, Zhang J, Chen Y, Rao L, Wang X, Zhao H, Wang B, Xiao Y, Yu J, Xu Y, Shi J, Han W, Song Z, Yu F. Small-Molecule Compound CY-158-11 Inhibits Staphylococcus aureus Biofilm Formation. Microbiol Spectr. 2023 May 11:e0004523. doi: 10.1128/spectrum.00045-23. Epub ahead of print. PMID: 37166296.

Yasir M, Dutta D, Hossain KR, Chen R, Ho KKK, Kuppusamy R, Clarke RJ, Kumar N, Willcox MDP. Mechanism of Action of Surface Immobilized Antimicrobial Peptides Against Pseudomonas aeruginosa. Front Microbiol. 2020 Jan 22;10:3053. doi: 10.3389/fmicb.2019.0305. PMID: 32038530; PMCID: PMC6987417.

Stiefel P, Schmidt-Emrich S, Maniura-Weber K, Ren Q. Critical aspects of using bacterial cell viability assays with the fluorophores SYTO9 and propidium iodide. BMC Microbiol. 2015 Feb 18;15:36. doi: 10.1186/s12866-015-0376-x. PMID: 25881030; PMCID: PMC4337318.

Question 7: 

Figure 7 does not present any statistical data, although it is indicated in the methodology that the experiment was performed in triplicate.

Response 7:

Sorry for our mistake. We have redrawn Figure 7 and the statistical data have been added, please check our revised manuscript.

Question 8:

The discussion is too superficial regarding the bioactivity results.

Response 8:

We have discussed the results of the biological activity in more depth in the Discussion section.

Reviewer 2 Report

In this study, 4,528 endogenous peptides were identified from 1,224 precursor proteins using peptidomics and screened for bioactivity in the buccal glands of the lamprey, Lethenteron camtschaticum. Four bioactive candidate peptides have been selected and chemically synthesized to test their predicted bioactivity as antimicrobials, as anti-inflammatory or as vasodilatant compounds.

From the conceptual and the experimental point of view this is a very interesting and systematic study.

Abstract: the secondary structure was not determined experimentally. This should be stated. It was calculated/predicted.

 Introduction: very nice written

 Results: 2.1 It remains completely unclear also from the Materials and Methods section (4.2) how the peptide extraction  was carried out. There was used a kit from Biotree Biotech. But nothing is stated how this kit separates effectively proteins and peptides, what is the molecular cut-off, which kind of beads are used, is it based on molecular size or hydrophobicity/hydrophilicity ? How was the recovery of the peptide fraction and how was carried out a kind of quality control of this exctration procedure ? How many peptides may have been lost during this procedure ? The protein concentration of the buccal secretion was estimated to be around 100 ug/ul. What was the peptide concentration of the peptide solution (2ul) applied to the NanoLC-MS/MS ?

I think these informations are imperative and must be added because this peptide exctraction step is the basis for the whole study. Figure 8 could be extended accordingly.

Line 283: 5-22% not 52-2%

I am fine with other presented data and the further Results section. It is missing information how these 4 candidate peptide sequences have been selected from a major list. You should show this list and describe the selection criteria for these 4 peptide sequences.

 In the Discussion´s section the added information about the peptide recovery obtained by this extraction method could then be discussed. In this context it could be discussed if the amount of 4528 peptide is  is a high value or not.

The Figure legends in the supplemental Figures S1 and S2 are missing.

Author Response

Reviewer 2

Abstract: the secondary structure was not determined experimentally. This should be stated. It was calculated/predicted.

Response:

Thank you for your suggestion. Already modified. 

Results: 2.1 It remains completely unclear also from the Materials and Methods section (4.2) how the peptide extraction  was carried out. There was used a kit from Biotree Biotech. But nothing is stated how this kit separates effectively proteins and peptides, what is the molecular cut-off, which kind of beads are used, is it based on molecular size or hydrophobicity/hydrophilicity ? How was the recovery of the peptide fraction and how was carried out a kind of quality control of this exctration procedure ? How many peptides may have been lost during this procedure ? The protein concentration of the buccal secretion was estimated to be around 100 ug/ul. What was the peptide concentration of the peptide solution (2ul) applied to the NanoLC-MS/MS ?

I think these informations are imperative and must be added because this peptide exctraction step is the basis for the whole study. Figure 8 could be extended accordingly.

Response:

Thank you for your suggestion. We have rewritten section 4.2 and rewrote the method of peptide extraction. Please check our revised manuscript. Regarding peptide loss, we lost about 20% of peptides due to ultrafiltration and desalination. For sample-saving consideration, we did not measure the peptide solution concentration. However, because the ultrafiltration filter has a concentrating effect, the concentration of the peptide solution applied to the NanoLC-MS/MS would be greater than 100 μg/μL.

We have also modified Figure 8 so that it now shows our work in more detail. 

Line 283: 5-22% not 52-2%

Response:

Already modified.

I am fine with other presented data and the further Results section. It is missing information how these 4 candidate peptide sequences have been selected from a major list. You should show this list and describe the selection criteria for these 4 peptide sequences.

Response:

Thank you for your suggestion. We have added the selection standards for predicting bioactivity in Table 2. Please check our revised manuscript.

In the Discussion´s section the added information about the peptide recovery obtained by this extraction method could then be discussed. In this context it could be discussed if the amount of 4528 peptide is a high value or not.

Response:

We have improved the discussion section and relevant content has been added.

The Figure legends in the supplemental Figures S1 and S2 are missing.

Response:

Already modified.

Round 2

Reviewer 2 Report

I am fine with this revised version.

Please proofread carefully final version for wording errors.